# Is RATS Superior to VATS in Thoracic Autonomic Nervous System Surgery?

**DOI:** 10.3390/jcm13113193

**Published:** 2024-05-29

**Authors:** Federico Raveglia, Angelo Guttadauro, Ugo Cioffi, Maria Chiara Sibilia, Francesco Petrella

**Affiliations:** 1Department of Thoracic Surgery, Fondazione IRCCS San Gerardo dei Tintori, 20900 Monza, Italy; maria.sibilia@unimi.it (M.C.S.); francesco.petrella@irccs-sangerardo.it (F.P.); 2Department of Medicine and Surgery, School of Medicine and Surgery, University of Milano Bicocca, 20126 Milan, Italy; angelo.guttadauro@unimib.it; 3Department of Surgery, School of Medicine and Surgery, University of Milan, 20122 Milano, Italy; ugocioffi5@gmail.com

**Keywords:** da Vinci, nerve reconstruction, hyperhidrosis, sympathectomy, compensatory hyperhidrosis, RATS

## Abstract

Technological development in the field of robotics has meant that, in recent years, more and more thoracic surgery departments have adopted this type of approach at the expense of VATS, and today robotic surgery boasts numerous applications in malignant and benign thoracic pathology. Because autonomic nervous system surgery is a high-precision surgery, it is conceivable that the application of RATS could lead to improved outcomes and reduced side effects, but its feasibility has not yet been thoroughly studied. This review identified three main areas of application: (1) standard thoracic sympathectomy, (2) selective procedures, and (3) nerve reconstruction. Regarding standard sympathectomy and its usual areas of application, such as the management of hyperhidrosis and some cardiac and vascular conditions, the use of RATS is almost anecdotal. Instead, its impact can be decisive if we consider selective techniques such as ramicotomy, optimizing selective surgery of the communicating gray branches, which appears to reduce the incidence of compensatory sweating only when performed with the utmost care. Regarding sympathetic nerve reconstruction, there are several studies, although not conclusive, that point to it as a possible solution to reverse surgical nerve interruption. In conclusion, the characteristics of RATS might make it preferable to other techniques and, particularly, VATS, but to date, the data in the literature are too weak to draw any evidence.

## 1. Introduction

Thoracic surgery is considered a challenging discipline because of the presence of important vascular structures, complex anatomy, difficult access to a narrow operating field, and severe postoperative pain. Against this backdrop, the last two decades have seen the worldwide advent of both multi- and uniportal VATS. This technique has been credited with introducing the concept of minimally invasive surgery in thoracic surgery. However, VATS is still considered challenging and characterized by a long learning curve [1] due to the absence of articulating devices and some critical technical issues, such as crushing of instruments in the narrow mediastinal space. Meanwhile, technological development of robotics has meant that, in recent years, more and more thoracic surgery departments have embraced this type of approach. In fact, the robotic platform has enabled increased expertise in dissection and reconstruction of delicate structures. Currently, numerous papers have demonstrated that the use of robotic assistance in thoracic surgery (RATS) can help optimize outcomes and maximize patient benefits in a variety of neoplastic and non-neoplastic diseases [2,3]. The advantages of RATS can be listed as follows: (1) improved vision thanks to the 10 times magnification and the depth perception of its 3D high-definition camera, (2) seven degrees of freedom (7DOF) allowing complex surgical maneuvers to be performed safely, reaching comfortably narrow spaces, (3) tremor filtering (6-Hz motion filter) and motion scaling systems that counteract physiologic hand tremors, allowing the surgeon to perform the finest surgical movements, (4) surgeon ergonomic position reducing fatigue and muscular struggle, (5) Firefly (fluorescence-capable vision system), which offers integrated high-resolution near-infrared fluorescence images in 3-dimensional vision, taking advantage of this technology to identify vascular anatomy. Regarding the learning curve, due to intuitive robotic movements, 25 RATS lobectomies, for example, seem to be sufficient for an experienced thoracic surgeon, unlike VATS [4]. In summary, robotic surgery has many applications in the chest, including benign and malignant esophageal surgery, mediastinal masses, and pulmonary lobectomy. Less common procedures include plication and reconstruction of the diaphragm and decortication of the pleura.

Thoracic autonomic nervous system surgery is based on the interruption of the central nervous system adrenergic effect; it currently finds its application in the treatment of essential hyperhidrosis and some cardiovascular diseases. Sympathetic block is widely performed for palmar, axillary, and facial hyperhidrosis [5]. Palmar hyperhidrosis is the main indication for nerve block, whereas axillary hyperhidrosis has been considered for surgery with the advent of VATS, also because medical therapy guarantees good results as well. Concerning facial hyperhidrosis, surgery is considered the last-line option because of frequent side effects. Nowadays, thoracic autonomic nervous system surgery has little space in cardiac pathologies; concerning arrhythmias, its role is still considered meaningful only in select patients. Nerve block is reemerging as a possible further therapy for ventricular resistant arrhythmias [6] in patients not responding to conventional drugs. The threshold to perform sympathetic block is somewhat lower for channelopathies (long QT syndrome) [7,8]. Recent pediatric-oriented guidelines [9] have recommended cardiac sympathectomy for ventricular tachycardia (VT) or VT/fibrillation storm refractory to antiarrhythmic medications, long QT syndrome, and catecholaminergic polymorphic VT. Sympathetic surgery is not indicated for vascular obstructive patterns, whereas a benefit has been proven in the case of Buerger’s disease [10]. Concerning spastic ischemia, sympathetic surgery may improve ulceration or gangrene symptoms in advanced stages [11]. Raynaud syndrome is the one that most benefits from sympathetic surgery in the advanced stages, but benefits are temporary [12]. As autonomic nervous system surgery is a precision surgery, given the small size and weakness of nerve structures, it is conceivable that the application of RATS could lead to improved outcomes and reduced side effects; however, its applicability has not yet been verified. Our aim is to provide an overview of the possible role of RATS in thoracic autonomic nervous system surgery, focusing on the following main topics: (1) the role of RATS in standard thoracic sympathectomy, (2) the role of RATS in selective procedures (ramicotomy), and (3) the role of RATS in nerve reconstruction.

## 2. Relevant Sections

### 2.1. The Role of RATS in Standard Thoracic Sympathectomy

As mentioned above, thoracic autonomic nervous system surgery is indicated in the treatment of both hyperhidrosis and some cardiac and vascular disorders. With the advent of thoracoscopic equipment, outcomes have greatly improved, especially regarding morbidity and mortality. For this reason, the use of minimally invasive techniques (endoscopic thoracic sympathectomy (ETS), a term used most often in the nonsurgical literature) in this field has long been standardized, albeit with different technical variations, especially about the approach to the thoracic cavity. Indeed, while the role of VATS is unanimously accepted, there is no definitive guidance on the use of a single- or multiportal approach [13] or the possible benefit offered by CO_2_ insufflation.

In 2008, the use of robotic technology for thoracic sympathectomy was first described for axillary and palmar hyperhidrosis [14]. The authors designed a prospective randomized trial enrolling 40 patients in 2 groups. In one group, a robotic camera support system, AESOP (Automated Endoscopic System for Optimal Positioning, Computer Motion Inc., Goleta, CA, USA), was used instead of traditional human camera support. All other surgical maneuvers were performed identically for both groups. Therefore, it cannot be considered a fully robotic approach, but a hybrid one. Safety criteria, such as surgical accidents, pain, and cosmetic results, and efficacy criteria, such as duration of surgery and camera use, anhidrosis, duration of hospitalization, compensatory hyperhidrosis, and patient satisfaction, were analyzed. Data analysis showed that a camera held by a robotic arm system is safe but less efficient than a human assistant holding the camera.

The description of a fully robotic-based approach has only occurred in the last decade. In 2021, during the 150th Annual Meeting of the German Society for Thoracic and Cardiovascular Surgery, Sandhaus et al. [15] presented a series of 24 patients undergoing RATS sympathectomy for hyperhidrosis. A one-stage bilateral approach was performed in 10 patients. The operative time was 106 ± 39 min. Material costs were estimated at 500 euros per case. There was no conversion and no hospital mortality. No complications occurred during the hospital stay. Follow-up evaluations showed that hyperhidrosis was successfully treated in all cases. The researchers concluded that this surgery can be performed safely and with good results. This single-center, single-arm, nonrandomized, retrospective, observational study has the merit of bringing attention to the use of RATS in hyperhidrosis management and encouraging further studies that may lead to definitive conclusions.

In 2023, two papers addressing the role of RATS in cardiac sympathetic denervation (CSD) were published. In February, Suwalski et al. [16] presented a case report describing their single experience with a case of type 2 LQTS with recurrent implantable cardioverter–defibrillator (ICD) discharges despite adequate beta-blocker (βB) treatment, in which, for the first time in Poland, bilateral RATS CSD was performed. The procedure was performed bilaterally using three 10-mm thoracoscopic ports in the 6th, 4th, and 3rd intercostal spaces (daVinci Xi robot, Intuitive Surgical, Mountainview, CA, USA); the sympathetic chain was identified and excised in the T2–T4 portion. At a 6-month follow-up, the patient was free of symptoms and complications. A few months later, Melinosky et al. [17] presented their study at the ISMICS 2023 Annual Meeting in Boston. They compared RATS and VATS (the standard approach) cardiac denervation in patients affected by ventricular tachyarrhythmia (VT) refractory to management with antiarrhythmic medications or cardiac ablation. A single-center retrospective study of all adult patients has been designed. The primary endpoint was to determine the effectiveness of RATS in reducing implantable cardioverter defibrillator (ICD) shock burden. From 2019 to 2021, a total of 34/67 patients underwent RATS cardiac denervation by a left triportal approach. The robot model is not reported. Sympathectomy from the inferior half of the stellate ganglion to the level of the fourth rib was performed by hook cautery. The authors compared short-term outcomes in the two groups, but statistical analysis has been summarily described. As it has been pointed out, the small number of the enrolled population likely prevented them from making a propensity score matching, diminishing the power of analysis. However, they found that both groups obtained an 89% reduction in ICD shocks in the first year after surgery, but RATS had a significantly shorter operative time and fewer postoperative complications (in particular pneumothorax). Intraoperative blood loss, median hospital stay, and follow up complication rates were similar (Table 1).

### 2.2. The Role of RATS in Selective Procedures (Table 2)

As mentioned, the use of VATS to perform sympathetic nerve block in the treatment of hyperhidrosis has long been standardized. In fact, despite some technical variations, success rates for palmar hyperhidrosis range from 95% to 97% [18,19,20], 60% to 80% for axillary hyperhidrosis [18,19], and about 75% for facial flushing [20]. But it is the rate of some dreaded surgical side effects that still causes concern, such as compensatory hyperhidrosis (reported in 50–97% of patients), gustatory sweating (32–49%) [19], and Horner’s syndrome (1–2.4%) [18]. Unfortunately, such wide ranges also depend on the patients’ subjective feelings and the surgeons’ definition. Some authors reported that the rate of compensatory hyperhidrosis could be lessened by only interrupting the T4 ganglion for palmar hyperhidrosis; however, this finding has not been validated in high-quality, large-scale randomized controlled trials [20].

To our knowledge, excellent results obtained in the management of compensatory hyperhidrosis have been reported by Friedel and colleagues [18,21,22,23] (2.5%) and were reached by adopting the selective sympathectomy surgical technique. Unlike ganglionectomy, this technique involves splitting the postganglionic efferent fibers (gray communicating branches), which branch from the second, third, and fourth sympathetic ganglia and head to the upper extremity [18,23,24]. Lee and colleagues [25] reported a prospective comparison of patients undergoing T2 sympathectomy and T3 ramicotomy for palmar hyperhidrosis, showing that compensatory hyperhidrosis was approximately two-thirds less frequent in the group undergoing T3 ramicotomy. In 2021, Vanaclocha et al. [26] published a prospective randomized study comparing VATS sympathicotomy and ramicotomy in patients affected by palmar hyperhidrosis. Twenty-one patients underwent T3–T4 sympathicotomy and 21 underwent T3–T4 gray ramicotomy; their endpoints were sweating resolution, quality of life at baseline and postoperatively, compensatory hyperhidrosis onset, postoperative hand dryness, patients’ satisfaction, surgical complications, and if they would undergo surgery again and recommend it. At 33 months mean follow-up, a postoperative quality of life scale analysis showed that patients undergoing sympathicotomy reported worse results (49.05 (SD: 15.66), IR: 35.50–63.00 vs. 24.30 (SD: 6.02), IR: 19.75–27.25). A higher figure in the quality-of-life scale means a smaller postoperative improvement. Indeed, patients in the sympathicotomy group sweated less in the hands, axillae, and forehead but much more in the abdomen, thighs, and feet. They also had a more significant postoperative temperature rise in the forehead, with a colder temperature in the abdomen, thighs, and soles of the feet than those in the ramicotomy group. This group also performed better regarding hand dryness and intention to undergo the procedure again or recommend it to somebody else. In conclusion, this randomized study suggested that grey ramicotomy had better results than T3–T4 sympathicotomy, with less compensatory sweating and higher patient satisfaction. Limitations were represented by the small number of patients enrolled and short-term follow-up, which may underestimate long-term recurrences. The rationale for the findings could be that chain division and ganglionectomy act by disrupting postganglionic sympathetic fibers that affect different and larger portions of the body than the upper extremity, while division of postganglionic sympathetic fibers that travel with intercostals 2, 3, and 4 ensures a more specific block directed at the upper extremity.

In 2013, Coveliers et al. [27] presented a case report of 55 patients (110 procedures) undergoing selective postganglionic thoracic sympathectomy with robotic technology for the first time. The rationale of their study was to verify the potential of selective sympathectomy (ramicotomy) when performed with the advantages of RATS, such as three-dimensional magnified high-definition vision and increased maneuverability in a confined space. They retrospectively reported a 5-year surgical experience of patients with persistent hyperhidrosis without apparent cause for more than 6 months despite medical management. Patients with palmar, axillary, and combined palmar and axillary hyperhidrosis were included in the study. They adopted a 4-port RATS approach to perform: (1) dissection of the pleura overlying the sympathetic chain from the second to the fourth rib, (2) division of the postganglionic branches from T2 to T4, (3) all accessory communicating fibers and disruption of Kuntz’s nerve by clearing the tissues lateral to the sympathetic chain. Surgery was always performed bilaterally at the same time. At a median follow-up of 2 years (range, 3–36 months), the overall resolution of hyperhidrosis was 96%, compensatory sweating was recorded in 7.2% of patients, and gustatory sweating in 1.8%. Complications observed included transient unilateral Horner’s syndrome (1.8%), unilateral dysesthesia of the right hand (1.8%), transient unilateral isolated ptosis (1.8%), and bradycardia (3.6%). Based on the encouraging results, the authors concluded that RATS selective sympathectomy may represent an effective, feasible, and safe procedure, with excellent relief of hyperhidrosis and low rates of compensatory sweating and complications. However, many limitations of this study (such as the absence of a control group and randomization, short follow-up, and small number of patients) prevent any firm conclusions from being drawn.

In 2020, Gharagozloo et al. [28] published another paper focused on robotic ramicotomy based on the assumption that the historic unsuccessful results may have been in part due to the limitations of the VATS visualization and instrument technology. They designed a retrospective data analysis of 47 patients who underwent two-stage RATS selective sympathectomy for axillary and palmar hyperhidrosis, within a group of 102 patients who were also operated on bilaterally at a single time. Their outcomes were postoperative sweating resolution, operative time, morbidity, death, compensatory hyperhidrosis, and gustatory sweating onset. They always adopted a two-stage RATS 3-port approach using both da Vinci (Intuitive Surgical Mountainview, CA, USA) Si and Xi robots. All the 47 patients underwent division of R2, R3, and R4 preganglionic and postganglionic rami. Regarding the treatment of hyperhidrosis and the occurrence of compensatory sweating, the overall resolution of hyperhidrosis was 98%. Compensatory hyperhidrosis was found in 19/47 (40%) patients after selective sympathectomy of the dominant upper limb. Contralateral dorsal sympathectomy was delayed until resolution of transient compensatory hyperhidrosis, which occurred within four weeks in all patients. Transient compensatory hyperhidrosis was observed in 21/47 patients (45%) after contralateral sympathectomy. This resolved in 46/47 patients within five weeks after surgery. At a mean follow-up of 28 ± 6 months, 98% of patients were free of prolonged compensatory hyperhidrosis. Compensatory hyperhidrosis involving the anterior abdomen and lower chest occurred in one patient (2%). No gustatory sweating was ever detected. Complications were transient heart block after second-sided sympathectomy in 2% and transient partial Horner’s syndrome that resolved in 2% of patients. No permanent Horner’s syndrome was recorded. These results were particularly encouraging, especially regarding complications and side effects, so the authors concluded that RATS ramicotomy (despite greater morbidity, longer operative times, and greater cost) is associated with excellent relief of hyperhidrosis and the lowest reported rate of compensatory hyperhidrosis (Table 2).

**Table 2 jcm-13-03193-t002:** Summary of the characteristics and outcomes of papers focused on the use of RATS in ramicotomy for hyperhidrosis.

Authors	Condition	Type of Study	Robotic Platform	N° Patients	Ramicotomy Extent	Results
Coveliers et al. [27]	Hyperhidrosis	case series analysis, retrospective	not reported	55 patients, 110 procedures	R2–R4	96% had relief of hyperhidrosis at a median follow-up of 24 months; compensatory sweating seen in 7.2%.
Gharagozloo et al. [28]	Hyperhidrosis	Single arm, retrospective	da Vinci Si and Xi	47 patients, 94 procedures	R3–R4	Relief of hyperhidrosis in 98% of patients. At 28 ± 6 months follow up 98% patients free of compensatory hyperhidrosis.

### 2.3. The Role of RATS in Nerve Reconstruction

Serious side effects or complications that make people regret sympathetic surgery and wish to cancel it are rare and mainly concern patients with hyperhidrosis. However, given the prevalence of severe focal hyperhidrosis, estimated at 3% of the population, there is a fair subgroup of patients whose quality of life is significantly worsened after surgery. Therefore, many authors prefer the clipping technique because of the presumed reversibility achievable with the removal of the clip itself [29]. The theory is that nerve compression may provide symptomatic efficacy, but also a hypothetically reversible procedure if the patient develops intolerable side effects. However, data in favor of reversibility in the real world are not encouraging [30]; therefore, some surgeons have explored the direct method of nerve reconstruction. Since the features of RATS are ideal for performing a microsurgical procedure (including high magnification, three-dimensional optics, scalability of motion, distal articulation, and microinstrumentation), some authors considered that the technical advantages of a robotic technique would allow nerve reconstruction to be successfully performed, and so they investigated its feasibility to reconstruct the thoracic sympathetic chain using an autologous interposition nerve graft and a direct suture.

In 2016, Connery reported a case series of 3 patients who underwent sympathetic nerve graft reconstruction using da Vinci from September 2010 to April 2013 [31]. All of them had previously undergone sympathectomy (between R2–R5 and R–R3 in terms of extent) for hyperhidrosis and complained of severe compensatory sweating and an inability to raise their heart rate. Two patients underwent nerve reconstruction using autologous intercostal nerve graft and direct terminal suture, while the last one underwent reconstruction with a sural nerve graft. The interval between sympathectomy and reconstruction ranged from 5 months to 4 years. In terms of outcomes, only the two patients who used the intercostal nerve and for whom the interval since the previous surgery was in the range of months had good results in terms of reduced sweating and improved quality of life.

In 2020, Chang et al. [32] reported their experience with 7 cases, one of which had already been described in a 2019 case report [33]. It was a retrospective (January 2017–May 2019) study that enrolled patients affected by severe compensatory sweating after ETS for face and or palmar hyperhidrosis. Previous nerve interruption ranged between R2–R3 and R2–R4, and the interval with reconstruction was between 2 and 25 years. They always adopted a 4 port robotic approach with the da Vinci Xi robotic cart, and the surgical steps were as follows: (1) identification of the healthy stumps of the proximal and distal sympathetic trunk and the second-to-fourth intercostal nerves, (2) harvest of the sural nerve to bridge the defect, and (3) sympathetic trunk defect reconstruction and the second-to-fourth intercostal nerves coaptation back, in an end-to-side fashion, with 9-0 nylon sutures. All the procedures were accomplished bilaterally at one time, without conversion or major complications. Remarkably, median operating time was 10.5 h. Unfortunately, the authors reported a follow-up of more than 6 months for only one patient who showed good improvement (change > 70% in a score from 1 to 10) in many areas of interest, such as dry hands, emotional health, anhidrosis above the nipple, and thermoregulation. The authors concluded that their paper was able to demonstrate that the procedure is feasible but should be considered only a pilot study.

In 2023, Chen et al. [34] published the largest series of patients undergoing robotic nerve reconstruction due to severe and intolerable compensatory sweating, expanding Chang’s series from 3 years earlier in terms of number of patients and duration of follow-up. From October to January 2021, they enrolled 23 patients. The operative technique was identical to Chang’s (4 ports, da Vinci Xi cart, sural nerve graft, both sides at one time). The follow-up, in addition to being based on the usual outcomes regarding quality of life, was enriched with a sauna test with thermography before and after surgery. Because of the number of patients, data analysis with the Wilcoxon matched-pairs signed-rank test was possible. Patient characteristics were uneven, as the level of previous ETS ranged from single T2 to T2–T9, and the length of the nerve defect from 6 to 13 cm. The procedure (mean operating time 8.5 ± 1.7 h) was confirmed to be feasible, without conversion and major complications. Notably, investigations at 6 months revealed that robotic reconstruction was effective in reducing the severity of compensatory sweating in all body districts. The observed improvements were not only effectively preserved but also continued to progress up to 24 months after surgery, highlighting the success of reinnervation of the sympathetic pathway. A similar trend was observed for thermoregulatory changes and gustatory hyperhidrosis. Therefore, the authors reinforced their own earlier conclusions about the feasibility of the technique, adding encouraging persistent data in terms of outcomes with up to 2 years’ follow-up.

In 2023, Rojas et al. [35] reported their experience in the surgical management of severe compensatory sweating after thoracic sympathectomy for hyperhidrosis. The purpose of their study was to investigate outcomes after nerve reconstruction and, most importantly, they compared RATS and VATS for the first time in the literature. In addition, a one-time evaluation of healthy volunteers (controls) was performed to validate the quality-of-life measures. This was a retrospective study based on prospectively collected data. They performed reconstruction by adopting a sural or an intercostal nerve graft. When they used the intercostal nerve, it was dissected immediately after distal sympathetic section, cut, and transposed with a rotation to the proximal sympathetic section. The nerve was then anastomosed end-to-end at the proximal end of the sympathetic nerve with 4 or 5 8/0 Prolene epineural sutures in RATS procedures. Biological glue was always applied to the anastomosis. In VATS procedures, anastomosis was achieved only with glue without any suturing. In cases where the robot was used, the authors adopted the da Vinci Xi platform through a 4-port approach for both intercostal nerve and sural nerve reconstruction indifferently. All patients underwent surgery for the contralateral side in the same way and for a second time. Between 2015 and 2021, they enrolled 14 patients. The mean time between the sympathectomy (ranging between T2 only and T2–T4) and nerve reconstructive surgery was 54 ± 13 months. VATS was performed in 10 cases and RATS in 4 cases. In 8 cases, they used the intercostal nerve, and in 6, the sural nerve. Six months’ follow-up was completed for all patients. The improvement in sweating was rated as excellent or good in 7 (50%), unchanged in 6 (43%), and worse in 1 patient (7%) at 6 months (Table 2). Seven out of the 8 patients (88%) with an intercostal nerve transfer reported an improvement in CH symptoms, whereas none of the patients with the sural nerve reported an improvement. Moreover, they found a significant difference favoring intercostal nerve compared to sural nerve use in postoperative DLQI, and no correlation between the VATS or RATS surgical approaches on univariate analyses. In comparing VATS and RATS, no differences emerged regarding primary (compensatory sweating management and quality of life improvement) and secondary outcomes (operative time, complication rate, length of stay). There were several limitations, such as the small population, the retrospective nature, the short follow-up, and the data being too heterogeneous. Therefore, the authors limited themselves to favorable conclusions regarding the feasibility of the technique, particularly robotic nerve micro-suture, and the validation of the tools for data collection and evaluation (Table 3).

## 3. Discussion

With high-definition visualization, 10 magnifications, and articulated micro-instrumentation within the thorax, the introduction of the robot should allow easier identification of the sympathetic trunk and, in particular, its ganglia and communicating branches, division of adhesions, and neurolysis, and allow anastomosis with the epineural suture. Not forgetting, however, that the latest generation of 3D VATS cameras offers a very good view of the surgical field [36]. However, further studies are needed to definitively compare 3D video-assisted thoracoscopic surgery to robotic surgery [37]. Based on these assumptions, our aim was to investigate the literature to assess whether there was already an application of this technology in clinical practice, and with what results. Regarding standard sympathectomy and its usual fields of application as management of hyperhidrosis and some cardiac and vascular conditions, the use of RATS is almost anecdotal. Arguably, the greatest reluctance to its use stems not so much from the feasibility and effectiveness of the technique as from the fear of some meaningful shortcomings, such as tactile feedback absence during dissection, the use of three or four 8 mm ports (in place of two), prolonged time, and higher costs. However, these limitations may ease in the future with cost reduction, also supported by refinement of the technique (e.g., use of only two robotic arms, as in VATS). In fact, the overall cost differences are mainly determined by the high consumable costs associated with RATS, and thus an improvement in technique leading to the use of only 1 or 2 instruments will result in cost savings by optimizing the cost-effectiveness of RATS [38]. A two-port technique could also reduce postoperative pain. Furthermore, regarding the perception of increased operative time, it is emerging that with proper training, pre-operative times, as well as intraoperative times, also decrease [39]. Lastly, let us consider that one of the reasons for failure of VATS sympathectomy, in terms of reduction of sweating and the onset of compensatory sweating, is the difficulty in correctly visualizing the many anatomical variations of the sympathetic nerve and its ganglia [40], so much so that most authors use the costal margin and not the ganglion as anatomical findings. RATS could exceed this limit. Instead, the impact of RATS may be decisive if we consider selective techniques such as ramicotomy. Ramicotomy is not currently considered a gold standard, despite some encouraging studies, because of weak evidence. In addition, selective division of postganglionic sympathetic fibers, when performed with conventional videoendoscopic techniques, can be challenging. The difficulty arises from the two-dimensional view and the limited maneuverability of the instrument. Thus, the still suboptimal results of some series of VATS ramicotomies, still affected by high compensatory sweating (67–95%) [41,42,43], may be due to the shortcomings of the conventional videoendoscopic technique and the consequent poor visualization of the anatomy of the sympathetic chain and communicating fibers.

In contrast, studies by Coveliers [27] and Gharagozloo [28] demonstrated the effectiveness of this technique when performed by RATS. Again, the key to success would lie in better vision of the individual communicating branches and greater precision in their dissection. Regarding sympathetic nerve reconstruction, there are several studies, although not conclusive, that point to it as a possible solution to reverse surgical nerve block in cases of severe side effects [44]. However, there are still many points to be clarified, such as the correct timing from the first surgery, the choice of nerve graft (sural or intercostal), the technique for anastomosis (glue or suture), and the timing for treatment of the opposite side. Again, the characteristics of RATS may make it preferable to other techniques and, particularly, VATS. However, few studies exist in the literature, with few patients enrolled and short follow-up. Moreover, these are difficult to compare with each other because they are very different in terms of the characteristics of the patients chosen, the surgical technique adopted, and the type of graft used. Rojas D. [35] is the only one to make a direct comparison between VATS and RATS, but based on very small numbers and without randomization. The only conclusions drawn by the different authors concern the feasibility of the technique and the possibility of performing a nerve suture, whereas this does not seem possible with VATS.

## 4. Conclusions

The applications of RATS in thoracic autonomic system surgery are manifold. While comparison with VATS in standard sympathectomy/sympaticectomy needs further investigation, its role in some emerging practices, such as ramicotomy, or in less common practices, such as nerve reconstruction, seems to be supported by the technical potential of the robotic platform. Future studies will tell us whether RATS can be a “game changer” in the treatment of diseases related to the autonomic nervous system and, particularly, hyperhidrosis.

## Figures and Tables

**Table 1 jcm-13-03193-t001:** Summary of papers focused on the use of a robotic technique in performing standard thoracic sympathectomy for any pathological condition.

Authors	Condition	Type of Study	Robotic Platform	N° Patients	Results
Martins Rua et al. [14]	Hyperhidrosis	randomized controlled trial	camera holder robotic system AESOP	38 divided in two groups	Total and surgical length was longer in robotic arm group (*p* − 0.001)
Sandhaus et al. [15]	Hyperhidrosis	Single-arm, retrospective.	not reported	13 patients, 24 procedures	Operating time: 106 ± 39 min, cost for robot-specific one-time material: 500 euro per case. No conversion, no in-hospital mortality. No complications. Hyperhidrosis successfully treated in all cases.
Suwalski et al. [16]	LQTS type 2	Case report	daVinci Xi robot, Intuitive Surgical, Mountainview, CA, USA	1	QTc shortened by 60 ms on a surface electrocardiogram to a value below 500 ms.
Melinosky et al. [17]	Ventricular tachyarrhythmia	Single-center, 2 arms (VATS vs. RATS), retrospective.	not reported	67	Shorter procedure duration, with a median of 129 min (*p* = 0.008), VATS more complicated by pneumothorax (*p* = 0.004) and overall complications (*p* = 0.01) compared with the RATS. At 1 year, both groups decreasing from a median of 4 to 0 shocks (*p* < 0.001); at 1 year, percentage of patients with persistent ICD shocks and the median of ICD shocks similar between two groups.

**Table 3 jcm-13-03193-t003:** Summary of the characteristics and outcomes of papers focused on the use of RATS in sympathetic nerve reconstruction with graft after sympathectomy.

Authors	Condition	Type of Study	Robotic Platform	N° Patients	Nerve Graft	Type of Reconstruction	Results
Connery D. [31]	Compensatory sweating + exercise intolerance	Case series	da Vinci SI	3	2 intercostal, 1 sural	Direct end to end suture	Heart Rate, Compensatory sweating, Quality of Life: Improved in 2/3 patients
Chang et al. [33]	Compensatory sweating	Case series	daVinci Xi robot, Intuitive Surgical, Mountainview, CA, USA	7	Sural	Sural bridge of the defect and 9-0 nylon sutures; second-to-fourth intercostal nerves were coapted back in an end-to-side fashion.	No follow-up but one case: 70% improvement of compensatory hyperhidrosis
Chen et al. [34]	Compensatory sweating	Prospective, single arm	daVinci Xi robot, Intuitive Surgical, Mountainview, CA, USA	23	Sural	Sympathetic trunk reconstructed using a sural nerve graft coapted to the involved intercostal nerve in a side-to-side fashion	At 6 months, CS decreased significantly at all body surface areas. Improvements maintained at 24 months. No evidence of recurrent hyperhidrosis.
Rojas et al. [35]	Compensatory sweating	Prospective, 2 arms, not randomized	4: daVinci Xi robot, Intuitive Surgical, Mountainview, CA, USA 10: VATS	14	6 sural, 8 intercostal	Intercostal nerve graft transposition, proximal suture, glue on anastomosis	No significant difference in outcomes between approaches.

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
