# Peer review of "Is RATS Superior to VATS in Thoracic Autonomic Nervous System Surgery?"

_jcm, 2024, doi:10.3390/jcm13113193_

Round 1

Reviewer 1 Report

Comments and Suggestions for Authors

The authors have presented an interesting study on the role of robotics (RATS) in thoracic autonomic nervous system surgery.

Herein my comments:

1) The first part of the 'Introduction' section the first part (line 31-64) should be shortened in lenght. 

2) Although the RATS approach may show undoubted advantages, it would be interesting to better evaluate the costs associated with using this method in comparison with VATS. This could be better explored in discussion section. 

3) In addition, it might be interesting to evaluate the outcomes in terms of duration of surgery compared with VATS and the incidence of complications.

4) Line 390-391. Please remove it.

5) I do not fully agree with the fact that robotics allows better vision than VATS. There are now very high-resolution 3D optics that allow a very good view of the surgical field. This discussion is probably a little dated.  

6) Probably given the length of the narrative review, the Authors should reduce the length of it. Also it should be interesting, given the Impact Factor of this Journal, to conduct a systematic literature review according to PRISMA guidelines. I think it would greatly improve the quality of this work. 

7) There is a duplication in references 2 and 22. Please correct.

Author Response

Response to reviewer 1

1) The first part of the 'Introduction' section the first part (line 31-64) should be shortened in lenght.

Ok, done

2) Although the RATS approach may show undoubted advantages, it would be interesting to better evaluate the costs associated with using this method in comparison with VATS. This could be better explored in discussion section.

Ok done at line 353

3) In addition, it might be interesting to evaluate the outcomes in terms of duration of surgery compared with VATS and the incidence of complications.

Ok done at line 356

4) Line 390-391. Please remove it.

ok

5) I do not fully agree with the fact that robotics allows better vision than VATS. There are now very high-resolution 3D optics that allow a very good view of the surgical field. This discussion is probably a little dated. 

I understand your objection even though we in our experience find the 3D robot view (truly immersive) more comfortable than the 3D VATS view. However, we have introduced a mention of your comment to the line 343

6) Probably given the length of the narrative review, the Authors should reduce the length of it. Also it should be interesting, given the Impact Factor of this Journal, to conduct a systematic literature review according to PRISMA guidelines. I think it would greatly improve the quality of this work.

Certainly, that of systematic review according to the PRISMA guidelines is a high-quality approach. However, given the very limited number of articles on the topic in the literature, we found our approach to be good enough.

7) There is a duplication in references 2 and 22. Please correct.

I have edited the references

Reviewer 2 Report

Comments and Suggestions for Authors

Dear Authors,

thank you very much for your well-written narrative review of the literature, investigating the role of robotics in thoracic autonomic nervous system surgery. Please pay attention to the following comments, pertaining to your manuscript:

1.      Line 172. Please correct: published a prospective randomized study comparing VATS

2.      Lines 383-384. Please correct as such: Moreover, this is difficult to compare with each other because they are very different in terms of characteristics of the patients chosen, the surgical technique adopted, and the type of graft used.

3.      Lines 389-390. Please delete: 5. Conclusions This section is not mandatory but can be added to the manuscript if the discussion is unusually long or complex.

4.      Please correct the word Authors as authors throughout your text.

Best Regards

Author Response

Point by point responses to reviewer 2.

 1   Line 172. Please correct: published a prospective randomized study comparing VATS

Done 

  1. Lines 383-384. Please correct as such: Moreover, this is difficult to compare with each other because they are very different in terms of characteristics of the patients chosen, the surgical technique adopted, and the type of graft used.

Done

  1. Lines 389-390. Please delete: 5. Conclusions This section is not mandatory but can be added to the manuscript if the discussion is unusually long or complex.

Done

  1. Please correct the word Authors as authors throughout your text

Done

Thank you for the favorable judgment and suggested corrections. all accepted and introduced into the text.